# Curcumin Facilitates Aryl Hydrocarbon Receptor Activation to Ameliorate Inflammatory Astrogliosis

**DOI:** 10.3390/molecules27082507

**Published:** 2022-04-13

**Authors:** Chun-Hua Lin, Chia-Cheng Chou, Yi-Hsuan Lee, Chia-Chi Hung

**Affiliations:** 1Department of Nursing, Kang-Ning University, Taipei 11485, Taiwan; 2School of Nursing, National Yang Ming Chiao Tung University, Taipei 112304, Taiwan; 3National Laboratory Animal Center, National Applied Research Laboratories, Taipei 115202, Taiwan; 1603019@narlabs.org.tw; 4Department and Institute of Physiology, National Yang Ming Chiao Tung University, Taipei 112304, Taiwan; yhlee3@nycu.edu.tw; 5Brain Research Center, National Yang Ming Chiao Tung University, Taipei 112304, Taiwan

**Keywords:** curcumin, aryl hydrocarbon receptor, astrogliosis, inflammation, ligand-receptor interaction, indoleamine 2,3-dioxygenase

## Abstract

Curcumin is an anti-inflammatory and neuroprotective compound in turmeric. It is a potential ligand of the aryl hydrocarbon receptor (AhR) that mediates anti-inflammatory signaling. However, the AhR-mediated anti-inflammatory effect of curcumin within the brain remains unclear. We investigated the role of AhR on the curcumin effect in inflammatory astrogliosis. Curcumin attenuated lipopolysaccharide (LPS)-induced proinflammatory IL-6 and TNF-α gene expression in primary cultured rat astrocytes. When AhR was knocked down, LPS-induced IL-6 and TNF-α were increased and curcumin-decreased activation of the inflammation mediator NF-κB p65 by LPS was abolished. Although LPS increased AhR and its target gene CYP1B1, curcumin further enhanced LPS-induced CYP1B1 and indoleamine 2,3-dioxygenase (IDO), which metabolizes tryptophan to AhR ligands kynurenine (KYN) and kynurenic acid (KYNA). Potential interactions between curcumin and human AhR analyzed by molecular modeling of ligand–receptor docking. We identified a new ligand binding site on AhR different from the classical 2,3,7,8-tetrachlorodibenzo-p-dioxin site. Curcumin docked onto the classical binding site, whereas KYN and KYNA occupied the novel one. Moreover, curcumin and KYNA collaboratively bound onto AhR during molecular docking, potentially resulting in synergistic effects influencing AhR activation. Curcumin may enhance the inflammation-induced IDO/KYN axis and allosterically regulate endogenous ligand binding to AhR, facilitating AhR activation to regulate inflammatory astrogliosis.

## 1. Introduction

Curcumin [1,7-bis-(4-hydroxy-3-methoxyphenyl)-1,6-hepta-diene-3,5–dione], a natural phenolic substance extracted from the perennial herb *Curcuma longa* (turmeric), has been used as a spice and food-coloring agent [1]. Numerous studies have demonstrated its various beneficial activities to human health and its potential as a therapeutic agent, including anti-inflammation [2], anti-oxidation [3], anti-coagulation [4], and anti-carcinogenesis [5]. Its anti-inflammatory effects have been demonstrated by reducing the expression of pro-inflammatory cytokines, including tumor necrosis factor (TNF-α), IL-1, -2, -6, -8, and -12, cyclooxygenase-2 (COX-2), and inducible nitric oxide synthase (iNOS) [6]. In the central nervous system (CNS), curcumin has also been reported to be effective against neuroinflammation, including the suppression of glial activation and pro-inflammatory chemokine/cytokine production in Alzheimer’s disease, ischemic stroke, and spinal cord injury [7,8,9]. Several mechanisms are involved in curcumin’s anti-neuroinflammatory effects, including the inhibition of NF-κB and MAPK signaling [10]. Further unearthing of the multivalent target of curcumin for its immune modulation in the brain is necessary.

In the brain, astrocytes are the most abundant neural cells and are responsible for maintaining brain homeostasis, providing nutrients to neurons, forming the blood–brain barrier (BBB), and participating in immune responses during brain injury or neurodegeneration [11]. Astrocytes can detect danger signals and switch from quiescence to reactivation, thereby responding to and modulating various inflammatory circumstances via the secretion of cytokines and chemokines to activate adaptive immune defense; this process is named astrogliosis and can be indicated by upregulation of glial fibrillary acidic protein (GFAP) [12,13,14]. However, depending on the timing and context during the pathological process, the role of astrogliosis may shift from alleviating to exacerbating inflammatory reactions and tissue damage [15]. Several signaling pathways are involved in inflammatory astrogliosis, in which NF-κB and STAT3 are the most dominant transcription factors that mediate pro-inflammatory and anti-inflammatory responses in reactive astrocytes, respectively. STAT3 supports astrocytes, serving as a protective phenotype, while NF-κB triggers damaging signaling [16,17,18]. Curcumin was found to inhibit NF-κB signaling in endotoxin-induced neuroinflammation in cultured microglia, thereby functioning against microglia-mediated neurotoxicity and enhancing neuronal survival [19]. Moreover, growing evidence indicates that curcumin can attenuate astrocyte reactivation in several CNS diseases [20]. However, the direct target of curcumin that mediates its anti-inflammatory action remains undetermined.

Aryl hydrocarbon receptor (AhR) is a ligand-activated transcription factor of the basic helix-loop-helix (bHLH) protein family and was reported to be a potential intracellular receptor for curcumin [21]. Its PAS-B domain has been demonstrated to have ligand-binding pockets [22]. AhR ligands are categorized as being from environmental pollutants, such as TCDD, polycyclic aromatic hydrocarbons (PAHs) [23], or from endogenous metabolites such as tryptophan-derived 6-formylindolo[3,2-b]carbazole (FICZ) [24] and kynurenine (KYN), kynureinic acid (KYNA) [25], and indoxyl sulfate (IS) [26]. In addition, some flavonoid substances, including α- and β-naphthoflavone (ANF and BNF), can bind to AhR as well [27]. The AhR pathway regulates a variety of cellular functions, including metabolism, proliferation, differentiation, and immunosuppression [28,29,30]. The ligand-activated AhR interacts with aryl hydrocarbon receptor nuclear translocator (ARNT) and then translocates to the nucleus where it binds to dioxin response element (DRE) on the promotor or enhancer regions of its target genes, leading to the gene expressions for the AhR functions such as xenobiotic metabolic enzymes *cyp1a1* and *cyp1b1*, immune modulators *Cd36* and *Irf8*, and cell cycle-related gene *E2F* [31]. Furthermore, AhR can be activated in lipopolysaccharide (LPS)-induced inflammation through the induction and activation of the tryptophan catabolizing enzyme indoleamine 2,3-dioxygenase (IDO), resulting in the production of AhR endogenous ligand KYN and KYNA [32]. Moreover, AhR regulates NF-κB signaling by interacting with it to mediate immunosuppression [33,34]. Further, AhR activation in astrocytes ameliorates autoimmune inflammation in multiple sclerosis [35,36]. However, while curcumin was reported to activate AhR, whether AhR mediates the beneficial effects of curcumin on astrogliosis-associated neuroinflammation remained unclear. 

To better understand the role of AhR in the action mechanism of curcumin on the inflammatory astrogliosis, we used AhR knockdown in astrocytes to unveil the curcumin-AhR causality. To further investigate the possibility of curcumin binding on AhR, we established an AhR molecular docking modeling with a narrowed grid. With this approach, we predicted and identified the binding situation of various AhR ligands on AhR. 

## 2. Results

### 2.1. Both Curcumin and AhR Mediated Anti-Inflammatory Responses in Astrocytes

To investigate the role of AhR in the immunomodulatory effects of curcumin in astrocytes with inflammation, we used bacterial endotoxin lipopolysaccharide (LPS) to induce inflammatory responses in primary cultured rat astrocytes. In our previous study, 1 µM of curcumin could attenuate LPS-induced proinflammatory chemokine in astrocytes [9]. Therefore, we used 1 µM of curcumin combined with 1 µg/mL of LPS to treat astrocytes and examined the expressions of TNF-α and IL-6 using quantitative PCR (qPCR). As shown in Figure 1A,B, LPS significantly induced IL-6 and TNF-α gene expression, and curcumin suppressed these effects. Curcumin alone did not affect these gene expressions.

AhR acts as an immunomodulator and was speculated to be activated by curcumin [37]. AhR knockdown was performed by the transfection of two AhR-specific siRNAs (siAhR) into primary rat astrocytes for 48 h. The knockdown efficiency of siAhR was confirmed to decrease the expression to 28% of the scrambled control (Figure 1C). LPS was applied to the cultured astrocytes at 48 h after the siAhR transfection. We found that AhR knockdown enhanced LPS-induced TNF-α mRNA levels (Figure 1D), indicating that AhR negatively regulates the proinflammatory response of reactive astrocytes. 

### 2.2. AhR Mediated Curcumin-Decreased NF-κB Activation in LPS-Stimulated Astrocytes

Based on the anti-inflammatory properties of curcumin and its potential receptor AhR (Figure 1), we further investigated whether AhR contributed to the anti-inflammatory mechanisms of curcumin. Since the major signaling pathways participating in the inflammatory astrogliosis are NF-κB p65 and JAK/STAT3 [38], we knocked down AhR in astrocytes and further investigated the activation of both pathways under curcumin and LPS treatments. Western blot analysis showed that the active NF-κB p65, phosphorylated at Ser 536 (pp65), was significantly increased by LPS in the scrambled group, while curcumin attenuated this induction (Figure 2A). Notably, in AhR-knockdown astrocytes, LPS also significantly enhanced pp65 levels, but curcumin could not attenuate this effect (Figure 2A). This suggests that AhR is involved in the inhibitory effect of curcumin on NF-κB activation in LPS-stimulated astrocytes. However, curcumin did not affect LPS-induced phosphorylation of STAT3 at tyr705 (pSTAT3), the JAK phosphorylation site, in astrocytes, and there was no difference between the groups with or without AhR-knockdown (Figure 2B). Furthermore, curcumin alone did not affect the activation of NF-κB and STAT3 (Figure 2A,B). Therefore, these data suggest that curcumin may reduce LPS-induced NF-κB activation in an AhR-dependent manner and curcumin does not influence the LPS-activated JAK/STAT3 pathway. 

### 2.3. Curcumin Enhanced LPS-Induced AhR Activation and Attenuated Astrogliosis

Based on the AhR-mediating curcumin effects on attenuating NF-κB activation and TNF-α gene expression in astrogliosis, we further examined whether AhR expression and activity would be influenced by curcumin, thus dampening the inflammatory response. We first examined the expression of AhR in cultured astrocytes treated with LPS in the presence or absence of curcumin. Western blotting showed that curcumin alone did not affect the protein expression of AhR. However, LPS significantly elevated AhR protein level whereas curcumin could not change this effect (Figure 3A). Interestingly, curcumin upregulated the LPS-induced AhR mRNA expression despite the decrease of AhR mRNA in curcumin treatment alone (Figure 3B). We further examined the expression of the AhR target gene CYP1B1 and extrapolated the curcumin-mediated AhR activity. The data showed that curcumin enhanced the LPS-induced CYP1B1 mRNA level from 1.9 to 2.7-fold in astrocytes, but curcumin alone did not affect CYP1B1 expression (Figure 3C). Since curcumin elevates LPS-induced AhR activity, we examined whether AhR nuclear translocation, indicating its active form upon ligand binding, would be increased by LPS and whether this effect would be promoted by curcumin. Immunostaining was performed using the antibodies against AhR and GFAP (a reactive astrocyte marker). The nucleus was indicated by DAPI staining. We found that LPS induced the translocation of AhR into the nucleus (Figure 3D). Notably, curcumin alone did not induce AhR translocation, but can enhance the LPS-increased AhR nuclear translocation (Figure 3D, upper panel). This is similar to the results pertaining to CYP1B1 mRNA (Figure 3C), suggesting that curcumin enhances LPS-induced AhR activation in astrocytes.

Next, GFAP staining was performed to observe astrocyte morphology and reactivation. Confocal imaging revealed that LPS induced astrocyte hypertrophy, an astrogliosis morphology, compared to the cell morphology in the control group, and that curcumin reversed this morphological change (versus that of cells treated only with LPS; Figure 3D). Curcumin alone showed a cell morphology similar to that of the control cells. These results suggest that curcumin enhances LPS-induced AhR activation, accompanied by attenuating astrocyte reactivation.

### 2.4. Curcumin Enhanced LPS-Induced Indoleamine 2,3-Dioxygenase (IDO) Expression

We further elucidated the possible mechanism of how curcumin affects AhR activation in astrocytes under inflammation. Here, we sought to examine whether curcumin influences IDO-KYN/KYNA-AhR metabolic cycle triggered by LPS stimulation. Results showed that LPS increased IDO mRNA expression, and curcumin further enhanced LPS-induced IDO gene expression whereas curcumin alone did not affect IDO expression by qPCR analysis (Figure 4). Thus, this result indicates that curcumin may facilitate the IDO-KYN/KYNA pathway by enhancing LPS-induced IDO expression. 

### 2.5. Curcumin and KYN/KYNA Occupied Distinct Ligand Binding Sites on Human AhR

IDO upregulation is mainly regulated by LPS-triggered NF-κB; in addition, AhR activation also forms a positive-feedback loop to contribute to its increase [39,40]. Curcumin can enhance LPS-induced AhR activation and IDO expression (Figure 3C,D), raising a possibility that curcumin may bind to AhR. To address this question, we used molecular docking modeling to predict the binding affinity of curcumin on AhR PAS-B domain. We chose several representative ligands of AhR, including environmental pollutants TCDD and 3MC, and endogenous ligands FICZ, KYN, and KYNA, for molecular docking analyses to compare their molecular conjugation with curcumin. The results demonstrated that TCDD and 3MC bound to a classical ligand-binding pocket of AhR PAS-B with a binding energy of −2.61 and −0.26 kcal/mol, respectively (Figure 5A). Surprisingly, tryptophan metabolites FICZ, KYN, and KYNA are dominantly bound onto a previously unknown binding pocket. The binding energy with AhR PAS-B is −0.15, −3.62, and −4.69 kcal/mol in FICZ, KYN, and KYNA, respectively (Figure 5B). These results indicate that there are two possible ligand-binding pockets on the PAS-B domain of AhR, docking for different types of ligands. Curcumin was predicted to dominantly dock onto the TCDD binding site of human AhR and extend to a new binding site with a binding energy of −3.79 kcal/mol, similar to a bridge crossing between both binding sites (Figure 5C). However, CH233191, a ligand-selective AhR antagonist had a binding energy of −2.33 kcal/mol with AhR PAS-B, and the binding pattern was similar to that of curcumin (Figure 5D). The data yielded a molecular docking prediction according to which different category ligands of AhR may bind onto their distinct binding sites. 

### 2.6. Co-Existence Model of Curcumin and Inflammation-Producing KYNA on AhR

Since curcumin and KYN/KYNA are predicted to fit to different ligand-binding pockets on the AhR PAS-B domain, we further investigated whether it could co-exist on AhR via binding onto their respective pockets. We analyzed the effects of different binding sequences of AhR ligands on their docking possibility, including KYNA and curcumin (Figure 6A), as well as KYN and curcumin (Figure 6B). According to the molecular docking results, when curcumin binds on the classic binding site first, KYNA and KYN had the potential to dock on the novel binding site on AhR (−2.63 kcal/mol binding energy between curcumin and KYNA, Figure 6A left; −3.09 kcal/mol binding energy between curcumin and KYN, Figure 6B left). Similarly, when KYNA or KYN binds onto the novel binding site, curcumin can fit close to the classic binding site (−6.82 kcal/mol between curcumin and KYNA, Figure 6A right; −2.88 kcal/mol between curcumin and KYN, Figure 6B right). Thus, when KYNA is pre-binding onto AhR, curcumin enters its binding pocket more easily due to a resultant decrease in curcumin’s binding energy. In this situation, curcumin and KYNA can form three H-bonds (curcumin_O2-KYNA_O2, 4.2 Å; curcumin_O4-KYNA_O2, 3.0 Å; curcumin_O2-KYNA_O3, 4.5 Å) to increase their binding stability (Figure 6C). Next, we chose 3MC to analyze the possibility of its coexistence with curcumin onto AhR for comparison with the results of curcumin-KYNA or curcumin-KYNA coexistence, because 3MC occupies the same ligand binding pocket with curcumin. The modeling result demonstrated that curcumin pre-occupies the classical binding site of AhR, resulting in an elevated binding energy of 3MC (22.84 kcal/mol), thereby being unfavorable for 3MC binding on AhR (Figure 6D left). Instead, when 3MC preoccupies the ligand-binding pocket, curcumin could bind to the novel ligand binding site according to the predicted binding energy (−1.82 kcal/mol, Figure 6D right). 

In sum, the molecular modeling results suggest that curcumin can coexist with KYN/KYNA on AhR by binding to the classical and novel ligand binding pockets, respectively. Whereas curcumin appears to compete with 3MC, which might interfere with the 3MC-induced AhR activation.

## 3. Discussion

Curcumin has anti-inflammatory properties and reportedly regulates the immunomodulator AhR, which potentially mediates neuroinflammation in the brain [41]. However, the mechanisms by which curcumin regulates AhR remain controversial [42]. Herein, we demonstrated that the protective effect of curcumin in the context of inflammatory astrogliosis involves the AhR pathway. Remarkably, we observed that curcumin regulated AhR towards anti-inflammation, likely by enhancing LPS-induced endogenous AhR ligands KYN and KYNA, and by simultaneously binding on AhR, ultimately regulating AhR activation. Based on our molecular docking prediction, the previously identified ligand-binding site mainly interacts with environmental AhR ligands, such as TCDD and 3MC (A site). A newly found ligand-binding site was dominantly occupied by endogenous tryptophan-metabolites, such as KYN, KYNA, and FICZ (B site). Intriguingly, curcumin is predicted to occupy the A site and can co-exist with the B site ligands. This finding is critical as it offers a new perspective regarding AhR activation and the effects of its ligands and modulators. A summarized diagram indicating the proposed mechanism of how curcumin-mediated AhR activation modulates inflammatory signaling in astrocytes during neuroinflammation (Figure 7). 

AhR appears to act as a brake on the LPS-induced proinflammatory cytokine TNF-α expression to modulate the inflammatory response (Figure 1); thus, AhR may play a vital role in astrogliosis during neuroinflammation. Curcumin can attenuate inflammatory NF-κB signaling activation via AhR in LPS-stimulated astrocytes. TNF-α and IL-6 are the target genes of NF-κB, and it is known that AhR interacts with the transcription factor NF-κB to modulate downstream cytokines and chemokines [43,44]. These results support our finding regarding AhR involvement in curcumin-mediated inflammatory astrogliosis through the NF-κB pathway. However, another major astrogliosis signaling pathway STAT3 was not affected by curcumin or AhR (Figure 2). Since STAT3 activation serves a hallmark of neuroprotective astrocyte phenotype, this result implies that curcumin preserves this beneficial signaling [17,18]. A recent study reported that AhR mediates STAT3 activation via its E3 ligase activity [45]. Nevertheless, whether curcumin attenuates the inflammatory response in astrocytes through co-existence with KYN or KYNA on the distinct binding sites of AhR, thereby modulating its activity for NF-κB interaction, worth further investigation.

LPS-induced AhR activity, as indicated by its target gene CYP1B1 gene expression, was enhanced by curcumin after 12 h treatment (Figure 3). The LPS-induced high expression of AhR has been reported in macrophages and microglia, mainly through triggering NF-κB activation, to bind on the AhR promoter, thereby driving AhR gene expression [37]. Although curcumin inhibits NF-κB activation, other transcription factors activated by curcumin may also contribute to the induction of AhR transcription. The nuclear factor erythroid 2-related factor (Nrf2) signaling pathway can be activated by curcumin [46,47], and may induce the formation of the transcription factor complex Nrf2–Jdp2–AhR to increase AhR promoter activity [48]. Additionally, the upregulation of AhR protein also contributes to LPS- and curcumin-mediated AhR activation (Figure 3). Notably, LPS increases AhR mRNA and seems to be more than its effect on AhR protein level, which could be due to the rapid degradation of the AhR protein by the ubiquitin proteasome pathway after it is activated [49]. The AhR mRNA level seems to reflect the demand of AhR via feedback from AhR activation. How exactly curcumin regulates AhR gene expression requires further research.

Our study was the first to reveal that AhR may have two distinct ligand-binding sites. This property may result in different cellular responses for various ligands via the regulation of protein–protein interactions and DNA–binding activity. It has been reported that different AhR agonists induce opposing effects on T cell differentiation, illustrating that ligand-binding status may influence gene transcriptional activities [50]. FICZ, but not TCDD for example, can make the activated AhR interact with Nrf2, a transcription factor with antioxidant defense function, to transcript several enzymatic antioxidant genes such as glutathione S-transferase M1 (GSTM1) and heme oxygenase-1 (HO-1) [51]. Interestingly, curcumin is reported to be an AhR ligand, but it is still controversial as an agonist and an antagonist [42,52]. Although we demonstrated that curcumin could not induce AhR activity on its own, it can selectively enhance LPS-induced AhR activity (Figure 3C). When astrocytes are exposed to inflammatory stimuli, the upregulation of IDO may increase the production of KYN and KYNA to interact with AhR for its activation. Meanwhile, curcumin treatment can increase IDO, which presumably generates KYN and KYNA. If curcumin binds to AhR with KYN or KYNA (Figure 6), it is possible that this co-existence would synergistically regulate AhR activity. Further validation is required to confirm whether curcumin regulates AhR activation via this dual binding site model.

Previous reports indicated that human AhR recognized the TCDD, PAH, and tryptophan-derived FICZ using the same classical ligand-binding site [53,54]. This may result from the setting using the demand of large-scale virtual screening, as most studies use border grid spacing (1 Å). Herein, we used AutoDock as the major docking tool and utilized a thinner grid spacing of 0.375 Å to reduce the chance of missing some docking possibilities and to search the binding area more thoroughly [55]. In addition, the ligands seem to depend on their structures, such as benzo-based or indole-based, to recognize the different binding sites. Thus, the present molecular model increases the reliability, leading to the identification of a potentially novel ligand binding site on AhR. 

With the molecular modeling used to predict the possibility of curcumin binding to AhR, we found that curcumin seems to have a high binding affinity for AhR (Figure 5C). One possibility is that the structure of curcumin is similar to the AhR antagonist CH233191 to bind on the classical ligand binding site and cross to the new identified ligand binding site (Figure 5E). Additionally, curcumin appears to bind on both sites of AhR depending on the different ligand coexistence (Figure 6). Curcumin appears to more favorably bind to the site of environmental ligands by molecular docking modeling and may co-exist and synergize with the ligand KYN, which is predicted to bind to the site of tryptophan-derived ligands (Figure 5 and Figure 6). Curcumin can stabilize KYNA in the binding pocket through additional H-bonds (Figure 6). Thus, AhR ligands may, in a coordinated fashion, regulate AhR-mediated signaling, such as curcumin and KYNA or KYN. Interestingly, it also seems to work as an AhR antagonist for the environmental ligand 3MC (Figure 6D). This result is similar to a finding that curcumin suppressed TCDD-induced DNA-binding activity of the AhR/ARNT heterodimer and metabolic enzyme activities [52], supporting the notion that curcumin may be an antagonist of environmental ligands. Herein, we provided a possible ligand-molecule binding model to explain the double-edged role of curcumin in regulating AhR activation. Nevertheless, further studies are needed to assess whether curcumin may serve as an antagonist for ligands of the classical binding site and become a coactivator for ligands of the discovered novel binding site.

## 4. Materials and Methods

### 4.1. Primary Culture of Astrocytes

Animal care and experimental procedures were performed according to the guidelines approved by the appropriate Institutional Animal Care and Use Committee (IACUC). Astrocyte cultures were prepared as described previously [38]. Cerebral cortices were collected from 2-day-old Sprague Dawley (SD) rats and were homogenized via mechanical dissociation. The cell suspension was diluted in DMEM/F12 (Thermo Fischer Scientific, Waltham, MA, USA) containing 10% heat-inactivated FBS, then cells were seeded at a density of 2 × 10^6^ cells in 75-cm^2^ flasks. After 10–14 days in culture, microglia and oligodendrocytes were removed by orbital shaking. The adherent astrocytes were subcultured and seeded in 24-well plates with 10% FBS-containing DMEM/F12 medium for 24 h. Astrocyte cultures that reach 80–90% confluence were used in the experiments.

### 4.2. Western Blot Analysis

Cultured astrocytes were lysed using a lysis buffer [50 mM Tris, 150 mM NaCl, 1% Triton-X, 0.5% SDS, 1 mM Na_3_VO_4_, phosphatase inhibitor cocktail I (Sigma-Aldrich, St. Louis, MI, USA), and a protease inhibitor cocktail (GE Healthcare, Chicago, IL, USA), pH 7.4]. The protein content of the lysates was measured. Protein extracts (50 μg per condition) were then loaded onto 7.5% sodium dodecyl sulfate-polyacrylamide gels (SDS-PAGE), electrophoresed, and then transferred onto polyvinylidene fluoride (PVDF) membranes (Aucklandsham, Buckinghamshire, UK). The membranes were then incubated with anti-AhR (Biomol Gm, Plymouth Meeting, PA, USA, 1:1000) and anti-glyceraldehyde-3-phosphate dehydrogenase (GAPDH; Biogenesis, Poole, UK, 1:5000) antibodies, and then were further hybridized with horseradish peroxidase (HRP)-conjugated secondary antibodies (Jackson ImmunoResearch Laboratories, West Grove, PA, USA, 1:10,000). Horseradish peroxidase (HRP)-reactive chemiluminescence reagents were used to reveal immunoreactivity (Pierce, Rockford, IL, USA). The intensity of each protein band was then analyzed and compared.

### 4.3. Small Interfering RNA Preparation and Transient Transfection

Small interfering RNAs (siRNAs) were purchased from Thermo Fisher Scientific (Waltham, MA, USA). The astrocytes were transfected with 40 pmol of either AhR-targeting siRNAs (5′-GCUGGACAAACUCUCCGUUtt-3′, targeting exon 2—XM_579375; 5′-GCAUUUUAAAUGAAGCCUAtt-3′, targeting exon 11—XM_579375) or a scrambled control RNA using Lipofectamine^TM^ 3000 (Invitrogen, Thermo Fischer Scientific, Waltham, MA, USA) in 24-well plates. The efficiency of AhR silencing was analyzed 72 h after transfection using Western blotting.

### 4.4. Real-Time Polymerase Chain Reaction (RT-PCR)

The RNA of the cultured astrocytes was extracted using a Trizol/phenol/chloroform buffer. The SuperScript II reverse transcriptase (Thermo Fischer Scientific, Waltham, MA, USA) was then used to obtain cDNAs. The expression of AhR, CYP1A1, CYP1B1, iNOS, TNF-α, and the internal control 18S was measured using RT-PCR (StepOnePlus™ Real-Time PCR System; Applied Biosystems, Foster City, CA, USA). Each measurement was performed at least in triplicate.

### 4.5. Immunofluorescence Labeling

The astrocytes were fixed with 4% formaldehyde in 20 mM PBS and then permeabilized with 0.5% Tween-20 in 20 mM PBS (0.05% PBST). The primary antibodies, anti-AhR, and anti-glial fibrillary acidic protein (anti-GFAP) were properly diluted (1:300) in 0.1% PBST and incubated with the fixed cells overnight at 4 °C. The anti-mouse-IgG Alexa Fluor^®^488 and anti-rabbit-IgG Alexa Fluor^®^594 (Thermo Fisher Scientific, Waltham, MA, USA) secondary antibodies were then incubated for 1 h at 25 °C. Images were acquired using a confocal laser scan microscope (Olympus FV-1000; Olympus, Tokyo, Japan); the digital images were merged using the Fluoview software (Olympus, Tokyo, Japan).

### 4.6. Protein Modeling

Protein modeling was performed using the Phyre2 server (Structural bioinformatics group, London, UK) [56] in four stages. HHblits was used to scan the protein sequence of the human AhR_PAS-B domain against the specially curated nr20 protein sequence database. The resulting multiple-sequence alignment was further used to predict the secondary structure using PSI-blast-based secondary structure PREDiction (PSIPRED, University College London Bioinformatics Group, London, UK). Of note, both alignment and secondary structure prediction were combined into a query-hidden Markov model. Then, results of the fold library scanned against a database of HMM proteins of known structure were used to construct the crude backbone-only models, while loop modeling and side-chain placement generated the model candidates. The generated models were then evaluated via the maximization of the confidence and coverage of the query sequence. PowerfUL CHain Restoration Algorithm (Pulchra) was used to build a complete skeleton of the final Cα structure, followed by side-chain addition using R3.

### 4.7. Molecular Docking

Molecular docking was performed using AutoDock 4.2.6 (Scripps Research, San Diego, CA, USA) [57]. The hAhR PAS-B model was selected as the docked model, and the 3D coordinate files of the ligands curcumin, 2,3,7,8-tetrachlorodibenzo-p-dioxin (TCDD), 3-methylcholanthrene (3MC), CH233191, kynurenine (KYN), and 6-formylindolo[3,2-b] carbazole (FICZ) were downloaded from the PubChem database [58]. Polar hydrogen atoms were added, and the Gasteiger charges were then computed using “AutoDockTools”; the coordinate files were transformed into the PDBQT format. Of note, AutoDock, which is based on the Lamarckian Genetic Algorithm (LGA), was used as a scoring function to predict the ligand-binding site and calculate the ligand-binding affinity; further, van der Waals interactions, hydrogen bonds, torsion terms, and electrostatic interactions were all accounted for. The docked model was first set rigid, and then docking compounds were allowed to adjust the bond and torsion angles. The grid size was set to 70, 50, and 50 along with the X-, Y-, and Z-axes, respectively, with a 0.375 Å grid spacing. The parameters used for the docking models were GA population size = 150, maximum number of energy evaluations = 2,500,000, and GA crossover mode = 2 points. One hundred docking runs were performed in each case. A root mean square deviation (RMSD) tolerance of 2 Å was chosen for the clustering of the docked structures.

### 4.8. Statistical Analysis

Statistical analysis was performed using GraphPad Prism^®^ 6 software (GraphPad Software, San Diego, CA, USA). Data were represented as the mean ± the standard error of the mean (SEM). Primarily, non-parametric one-way ANOVA was used in more than two groups for the comparison of data between groups, followed by Bonferroni posthoc analysis. Additionally, the unpaired *t*-test was used to compare specified group pairs. Statistical significance was defined as *p* < 0.05.

## 5. Conclusions

This study elucidated an anti-inflammatory mechanism for curcumin in astrocytes that required AhR mediation and a more detailed understanding of the mechanism of curcumin in terms of its therapeutic effects on neuroinflammation resulting from brain diseases. Curcumin can enhance inflammation-induced IDO expression, which could promote the production of the AhR ligand KYN to activate AhR. In addition, it has the potential to bind onto AhR to attenuate inflammatory responses in astrocytes. Even though curcumin is not able to activate AhR, it can bind together with inflammation-induced endogenous AhR ligands onto the different AhR ligand pockets. These molecular interactions may further contribute to enhancing AhR activation under inflammatory conditions, thereby executing its anti-inflammation process in astrocytes. Herein, we provided a novel perspective showing the presence of two ligand-binding sites on AhR, illuminating a new and better design for AhR agonists and antagonists, thereby prompting the clinical application of therapeutic AhR-targeting drugs. This system may experimentally help assess the possibility of various AhR ligands causing diverse cellular outcomes. This ligand and AhR docking system will be able to be used for precise AhR-target drug development.

## Figures and Tables

**Figure 1 molecules-27-02507-f001:**
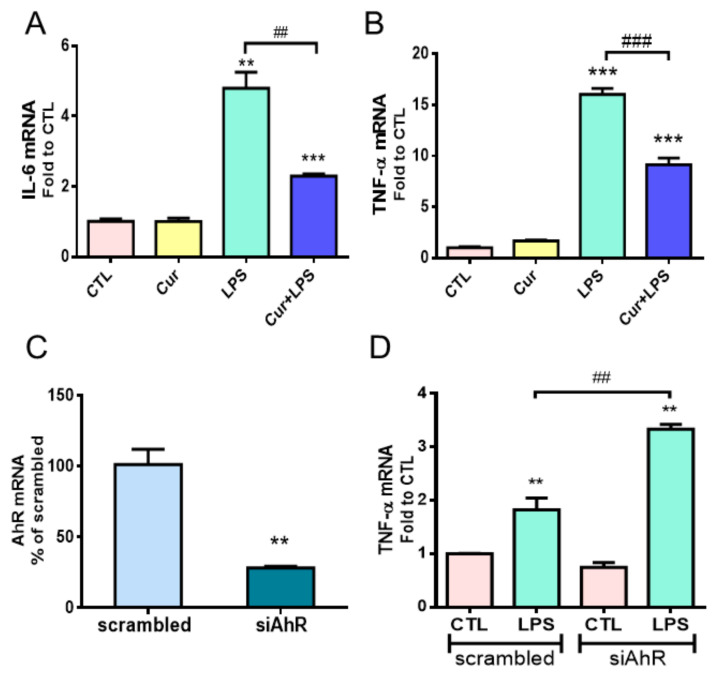
Effects of curcumin and AhR in astrocytes under inflammation. (**A**,**B**) Primary cultured rat astrocytes were treated with 1 µg/mL LPS with or without 1 µM curcumin (Cur) for 12 h. The mRNA expressions of IL-6 (**A**) and TNF-α (**B**) were analyzed by qPCR. Note that curcumin alleviated the LPS-induced inflammatory response in astrocytes. (**C**,**D**) Primary cultured astrocytes were transfected with AhR siRNA (siAhR) or scrambled RNA (as the control) for 48 h, followed by 1 µg/mL LPS treatment for 12 h. qPCR analysis of AhR mRNA expression indicated the knockdown efficiency of siAhR (**C**). Quantitative RT-PCR of TNF-α mRNA expression shown in (**D**). *n* = 3. Note that knockdown of AhR expression aggravated the inflammatory response of astrocytes under LPS stimuli. Statistical differences were analyzed using unpaired *t*-test and are represented as ** *p* < 0.01, *** *p* < 0.001 compared with the individual control group and ^##^
*p* < 0.01, ^###^
*p* < 0.001 compared with the indicated group using the unpaired *t*-test.

**Figure 2 molecules-27-02507-f002:**
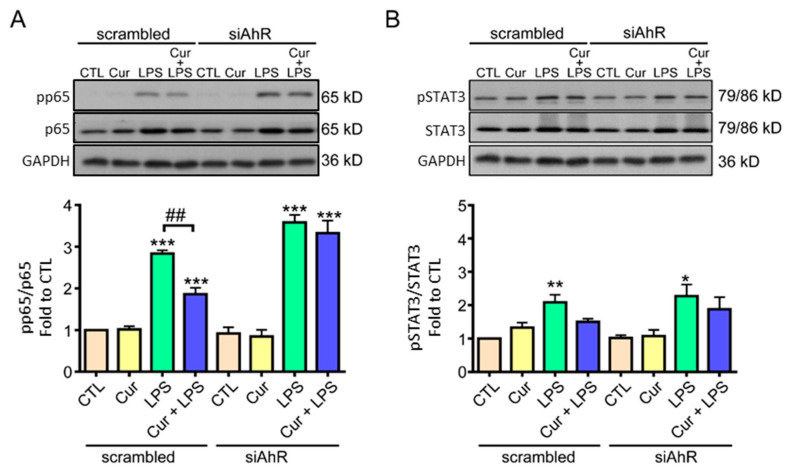
Curcumin decreased LPS-induced NF-κB activation, but not STAT3, via AhR. (**A**,**B**) Primary cultured rat astrocytes were transfected with scrambled siRNA (as the control), or si-AhR for 48 h and then were treated (or not) with LPS (1 μg/mL), curcumin (1 µM), or a combination of curcumin and LPS for 12 h. The levels of NF-κB pp65 (Ser536) and NF-κB p65 (**A**), as well as pSTAT3 (Tyr705) and total STAT3 (**B**), were determined by western blotting. Representative images and quantitative data are shown. Note that AhR knockdown blocked curcumin-reduced NF-κB activation in LPS-treated astrocytes, and STAT3 activation was not affected by curcumin. Statistical differences were analyzed using the unpaired *t*-test, and are represented as: * *p* < 0.05, ** *p* < 0.005, *** *p* < 0.001 compared with individual control group. ^##^
*p* < 0.01 compared with the indicated group.

**Figure 3 molecules-27-02507-f003:**
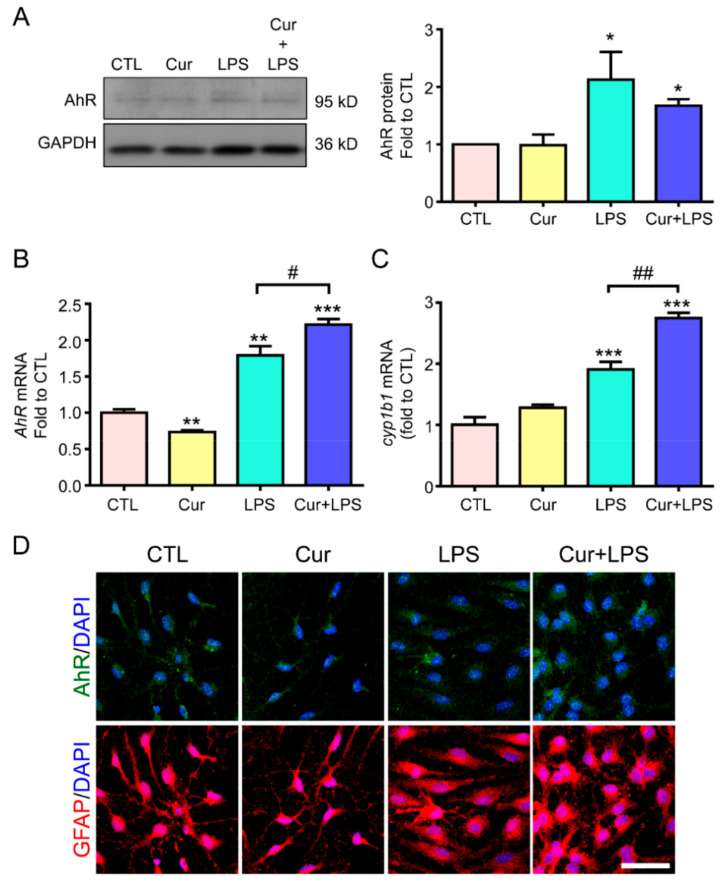
Curcumin enhanced LPS-induced AhR activation and nuclear translocation in astrocytes. Primary astrocytes were treated with 1 µg/mL LPS, 1 µM curcumin (Cur), or their combined treatment for 12 h in RNA and 24 h in protein. (**A**) AhR protein expression was determined by western blotting. GAPDH was used as the loading control. A representative blot and the quantification results are shown. *n* = 3. (**B**,**C**) AhR (**B**) and CYP1B1 (**C**) mRNA expression was determined by RT-PCR. Results are presented as fold expression differences over the control condition. (**D**) Immunofluorescence was performed to evaluate the expression of AhR (green, upper panel) and GFAP (an astrocyte marker; red, lower panel); DAPI (blue) was used to counterstain the nuclei. Representative images are shown. Note that curcumin enhanced LPS-induced AhR mRNA, not protein level, and elevated LPS-increased CYP1B1 mRNA expression and AhR nuclear translocation in astrocytes. Scale Bar = 20 μm. Statistical differences were analyzed using the unpaired *t*-test, and represented as: * *p* < 0.05, ** *p* < 0.005, *** *p* < 0.001 compared with the control group and ^#^
*p* < 0.05, ^##^
*p* < 0.01 compared with the LPS group. DAPI, 4′,6-diamidino-2-phenylindole.

**Figure 4 molecules-27-02507-f004:**
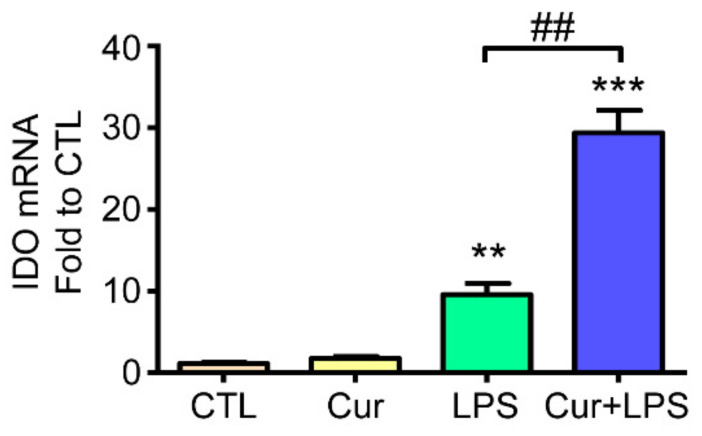
Effects of curcumin and LPS on IDO expression. Rat cultured astrocytes were treated with 1 µM curcumin, 1 µg/mL LPS, and their combined treatment for 12 h. Treated cells were harvested and analyzed regarding IDO mRNA expression by RT-PCR. Note that LPS induced IDO mRNA expression and curcumin further enhanced this effect. Data represent the mean ± SEM (*n* = 3). Statistical differences were investigated using the one-way ANOVA test and are represented as ** *p* < 0.01, *** *p* < 0.001 compared with the control group. ^##^
*p* < 0.01 compared with the LPS group.

**Figure 5 molecules-27-02507-f005:**
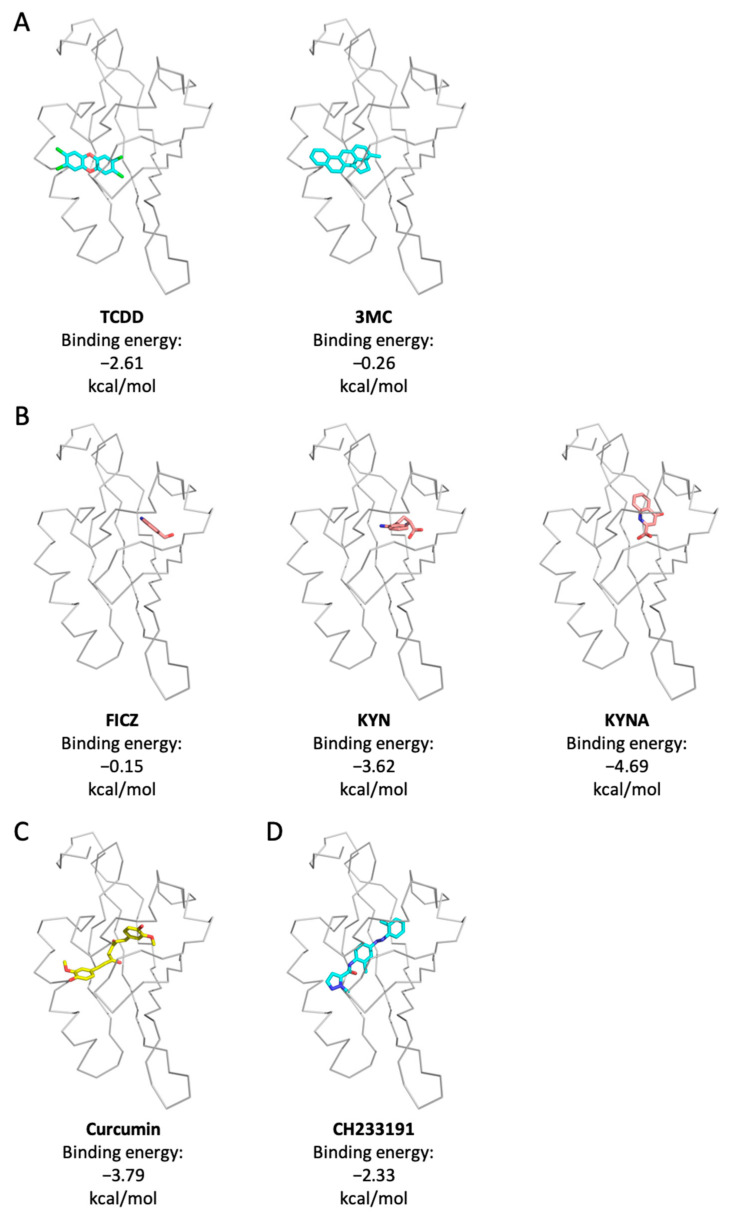
Molecular docking modeling of curcumin and classic AhR ligands with AhR. Binding pockets predicted for AhR PAS-B domain. Binding models of TCDD and 3MC (**A**), FICZ, kynurenic acid (KYNA), and kynurenine (KYN) (**B**), curcumin (**C**), and CH233191 (**D**) in the context of human AhR are shown. Note that TCDD and 3MC bound on the classic binding site of AhR, while FICZ, KYN, and KYNA bound on a newfound ligand binding site of AhR. Curcumin bound to the classical binding site. AhR, Aryl hydrocarbon receptor; TCDD, Tetrachlorodibenzo-p-dioxin; IDO, Indoleamine 2,3-dioxygenase; FICZ, 6-Formylindolo[3,2-b] carbazole.

**Figure 6 molecules-27-02507-f006:**
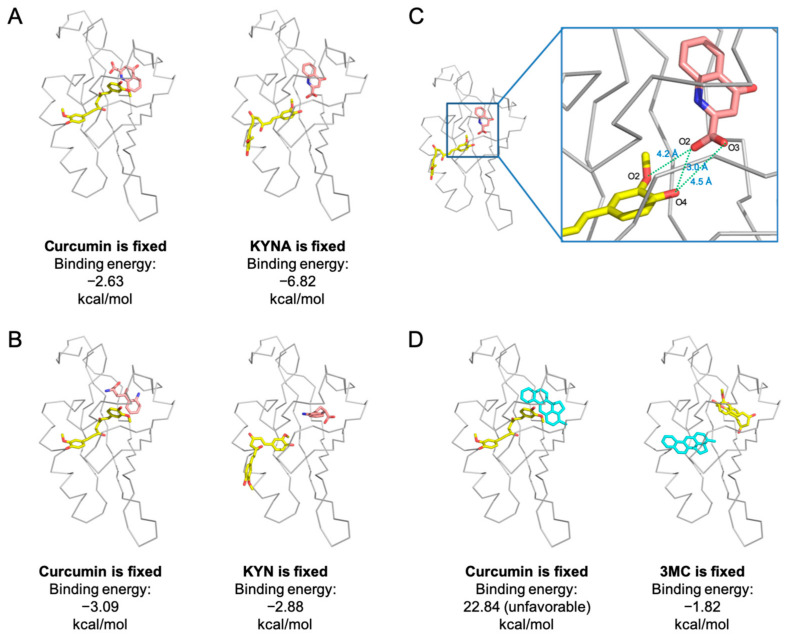
Curcumin and interactions of the bound ligands. AhR is represented in gray; the environment of the ligand-binding cavity is shown for the equilibrated model colored by curcumin in yellow, 3MC in blue, and KYNA and KYN in pink. (**A**) Modeling co-existence of curcumin and KYNA in different binding sequences. (**B**) Modeling co-existence of curcumin and KYN in different binding sequences. (**C**) Modeling studying AhR binding to curcumin and KYNA. The O2 and O4 positions of curcumin formed hydrogen bonds (green dash line) with the O2 and O3 of KYNA. (**D**) Modeling co-existence of curcumin and 3MC in different binding sequences. Note that the curcumin may be able to co-exist on AhR with KYNA or KYN, and may block 3MC binding onto AhR.

**Figure 7 molecules-27-02507-f007:**
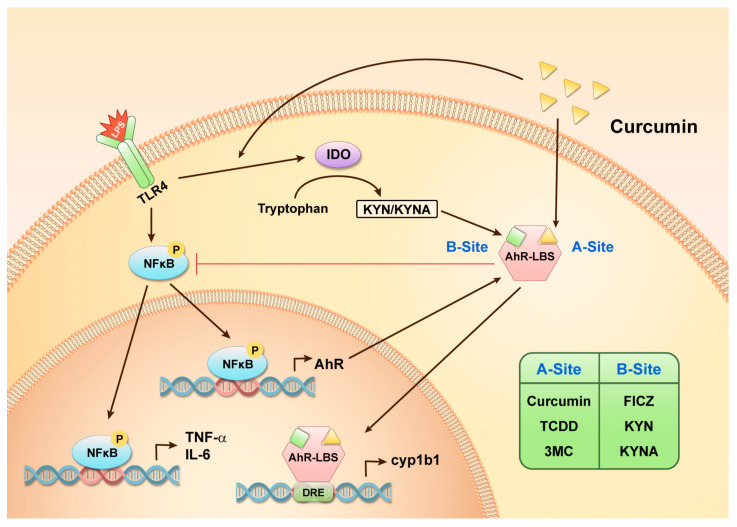
Mechanism of the anti-inflammatory effects of curcumin in the context of LPS-induced inflammation in astrocytes. The figure illustrates an overview of the curcumin-AhR pathway, which is involved in regulating LPS-induced proinflammatory cytokines. Our findings indicated that AhR may have two ligand-binding sites and these different ligands may synergistically modulate LPS-induced inflammation as shown in the figure. AhR-LBS, AhR ligand-binding site.

## Data Availability

The data generated and analyzed during the study are available in this article.

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
