# Peer review of "Curcumin Facilitates Aryl Hydrocarbon Receptor Activation to Ameliorate Inflammatory Astrogliosis"

_molecules, 2022, doi:10.3390/molecules27082507_

Round 1

Reviewer 1 Report

This manuscript demonstrates the anti-inflammatory mechanisms of curcumin in astrocytes and supports its therapeutic effects in neuroinflammation resulting from brain disease.

This is a good manuscript on technically correct and summing up interesting results.

The manuscript is well written and presents useful methods for the analysis of the anti-inflammatory effects of curcumin on LPS-stimulated astrocytes.

I suggest some minor revisions.

The conclusions can be improved with more details and explanations of the results obtained.

Line 135: “.. astrocytes under LPS stimuli” can be changed with “LPS-stimulated astrocytes”.

Line 136: The sentence: “Since both curcumin and AhR can attenuate LPS-induced inflammation in astrocytes (Fig. 1), we investigated whether AhR contributed to the molecular mechanisms of the anti-inflammatory effect of curcumin.” is hard to read. Please consider rephrasing it.

Line 138: The expression “Owing to the fact that” can be changed with “Because” / “Since”.

Line 138: “On the basis of..” can be replaced with “Based on..”.

Reviewer 2 Report

The article entitled „Curcumin facilitates aryl hydrocarbon receptor activation to ameliorate inflammatory astrogliosis” authored by Chun-Hua Lin, Chia-Cheng Chou, Yi-Hsuan Lee, Chia-Chi Hung, concerns a significant subject. This article is in the scope of this journal and could be interesting for the readers of this journal. The used methods are selected correctly. The work is written very well linguistically correct. The article provides all necessary explanations. The presented report presents new information. A pervasive literature review - contains 71 references. Although the work includes many literature references, they are not up-to-date - only 32% of the references are from the last five years. It is also necessary to pay attention that the links meet the editorial requirements (bold, italics in the appropriate places).

Reviewer 3 Report

I have reviewed the manuscript titled "Curcumin facilitates aryl hydrocarbon receptor activation to ameliorate inflammatory astrogliosis" authored by Lin et.al.

Though the researchers have designed the work in an appreciable manner, I have some basic issues that needs to be incorporated.

  1. Dose response studies should be added to provide the explanation of choosing the 1uM conc. of curcumin for the whole study.
  2. Why combined effect of LPS and curcumin was evaluated after 12h only??
  3. English editing by native speaker is necessary to ensure that there are no issues related to syntax, language, and grammar.
